# Efficacy of Clinically Used PARP Inhibitors in a Murine Model of Acute Lung Injury

**DOI:** 10.3390/cells11233789

**Published:** 2022-11-26

**Authors:** Vanessa Martins, Sidneia S. Santos, Larissa de O. C. P. Rodrigues, Reinaldo Salomao, Lucas Liaudet, Csaba Szabo

**Affiliations:** 1Section of Science and Medicine, University of Fribourg, 1700 Fribourg, Switzerland; 2Division of Infectious Diseases, Department of Medicine, Escola Paulista de Medicina, Federal University of São Paulo (EPM/UNIFESP), São Paulo 04023, Brazil; 3Service of Adult Intensive Care Medicine, University Hospital Medical Center, Lausanne University, 1015 Lausanne, Switzerland

**Keywords:** cell death, inflammation, cytokines, olaparib

## Abstract

Poly(ADP-ribose) polymerase 1 (PARP1), as a potential target for the experimental therapy of acute lung injury (ALI), was identified over 20 years ago. However, clinical translation of this concept was not possible due to the lack of clinically useful PARP inhibitors. With the clinical introduction of several novel, ultrapotent PARP inhibitors, the concept of PARP inhibitor repurposing has re-emerged. Here, we evaluated the effect of 5 clinical-stage PARP inhibitors in oxidatively stressed cultured human epithelial cells and monocytes in vitro and demonstrated that all inhibitors (1–30 µM) provide a comparable degree of cytoprotection. Subsequent in vivo studies using a murine model of ALI compared the efficacy of olaparib and rucaparib. Both inhibitors (1–10 mg/kg) provided beneficial effects against lung extravasation and pro-inflammatory mediator production—both in pre- and post-treatment paradigms. The underlying mechanisms include protection against cell dysfunction/necrosis, inhibition of NF-kB and caspase 3 activation, suppression of the NLRP3 inflammasome, and the modulation of pro-inflammatory mediators. Importantly, the efficacy of PARP inhibitors was demonstrated without any potentiation of DNA damage, at least as assessed by the TUNEL method. These results support the concept that clinically approved PARP inhibitors may be repurposable for the experimental therapy of ALI.

## 1. Introduction

Poly(ADP-ribose) polymerase 1 (PARP1) is the major isoform of a family of ADP-ribosylating enzymes. It has been implicated in the regulation of various biological processes, including DNA repair, gene transcription, and cell death [1,2,3,4,5,6]. PARP1, as a potential target for the experimental therapy of acute lung injury (ALI), was identified over 20 years ago [7,8]. However, clinical translation of this concept was not possible due to the lack of clinically approved PARP inhibitors. With the clinical introduction of several novel, ultrapotent PARP inhibitors (with ovarian and breast cancer being their approved indications), the concept that PARP inhibitors may have potential clinical utility in various non-oncological diseases, including ALI, through drug repurposing has received renewed justification [9,10,11,12,13]. The current project evaluated the effect of the clinical-stage PARP inhibitors olaparib, veliparib, rucaparib, talazoparib, and niraparib in oxidatively stressed cultured epithelial cells and monocytes in vitro and also assessed the efficacy of two selected inhibitors, olaparib and rucaparib in a murine model of ALI in vivo. These data demonstrate the preclinical efficacy and safety of clinically approved PARP inhibitors in ALI and support the concept of therapeutic repurposing of olaparib for the experimental therapy of ALI.

## 2. Materials and Methods

### 2.1. Cell Culture

The human monocyte cell line (U937) and the immortalized human bronchial epithelial cell line, BEAS-2B, were purchased from ATCC (American Type Culture Collection, Manassas, VA, USA). U937 cells were maintained in RPMI1640 with 10% fetal bovine serum (Life Technologies, Carlsbad, CA, USA). BEAS-2B cells were grown on cell culture dishes coated with bovine serum albumin, fibronectin, and bovine type I collagen in a bronchial epithelial growth medium (BEGM; Lonza, Basel, Switzerland). Cells were grown at 37 °C in a humidified atmosphere of 5% CO_2_ and 95% room air with culture medium replaced every 2 days.

### 2.2. Cell Viability Assays

U937 cells and BEAS-2B cells were seeded in a 96-well plate at 50,000 cells/well in 200 μL of complete culture medium. After 24 h, a fresh complete medium was added, and cells were challenged with H_2_O_2_ (600 µM for U937 cells and 30 µM for BEAS-2B cells). The oxidant concentrations were selected based on prior concentration-response studies to identify oxidant concentrations that induce comparable degrees of damage in the two cell types. Based on these experiments, the endothelial cells were found to be substantially more sensitive to the oxidant than the U937 cells. After 1 h, various PARP inhibitors were added to the cells to achieve the final concentration of 1, 3, 10, or 30 µM, and the plate was incubated for 24 h at 37 °C and 5% CO_2_. The purpose of this “post-treatment” regimen was to model the clinical situation, where therapies for ALI would only be feasible after the diagnosis of ALI (as opposed to pre-treatment).

The activity of lactate dehydrogenase (LDH) released from necrotic damaged cells was measured to determine the degree of oxidative-stress-induced necrosis, and the effect of the PARP inhibitors on this response/LDH activity was analyzed in 50  μL of cell supernatant using the Cytotoxicity Detection Kit (LDH, Roche, Basel, Switzerland) according to manufacturer’s instructions. Absorbance was measured at 490  nm, with 680 nm used as background. On the adherent cells, the MTT assay was performed. For the assay 50 µL of 3-(4,5-dimethylthiazol-2-yl)-2,5-diphenyltetrazolium bromide (MTT) at 2 µg/mL in PBS was added to the cells and incubated for 3 h at 37 °C and 5% CO_2_. The product of the reaction (conversion of MTT to formazan dye) was then detected. Cells were dissolved in 150 µL of DMSO, and formazan was quantified by measuring absorbance at 590 nm, with 690 nm (background), using Infinite 200 Pro reader (Tecan). The MTT assay is considered a general viability assay that is considered either a mitochondrial/cellular metabolism activity assay or, more generally, a read-out of general cellular metabolic activity due to cellular oxidoreductase activity.

### 2.3. Quantification of DNA Damage In Vitro

The magnitude of DNA fragmentation in cultured cells was evaluated and quantified using an in situ quantitative colorimetric apoptosis detection system, “HT TiterTACS™ Apoptosis Detection Kit” (R & D Systems, Minneapolis, MN, USA, Catalogue # 4822-96-K). Briefly, cells were cultured and plated in 96-well plates to the desired confluence (2 × 10^4^ cells/well). Cells were then fixed with 3.7% buffered formaldehyde for 5 min, followed by washing with PBS. Then, the cells were permeabilized with 100% methanol for 20 min, followed by washing with PBS. Cells were then subjected to a labeling procedure following the manufacturer’s instructions; the 3′-hydroxyl nick-end-incorporated biotin-conjugated dNTP was detected with streptavidin-conjugated HRP using the HRP-substrate TACS-Sapphire as a chromophore for the colorimetric read-out. The reaction was stopped with 2 N HCl after 30 min of substrate addition, and the absorbance was measured at 450 nm with a microplate reader. For comparison, a positive control (nuclease-treated control) was kept to confirm the permeabilization and labeling reaction. Data are expressed as experimental blank-corrected O.D450 (λ450) values from three technical replicates for each of the five biological replicates belonging to each experimental group.

### 2.4. Animal Model of ALI

C57/BL6 mice (male and female, 8–10 weeks old) were purchased from Janvier Laboratories (Le Genest-Saint-Isle, France). They were housed in a light-controlled room with a 12-h light-dark cycle and were allowed ad libitum access to food and water. All investigations adhered to the Guide for the Care and Use of Laboratory Animals published by the National Institutes of Health (Eighth Edition, 2011) and were performed according to a protocol approved by the local cantonal authority (FR Service de la sécurité alimentaire et des affaires vétérinaires SAAV, Protocol cantonal number #2020-43-FR) and the animal care and use committee of the University of Fribourg, Switzerland.

For the induction of ALI, mice were anesthetized and instilled intratracheally (i.t.) with LPS (*Escherichia coli*; O127:B8) (50 µg in 30 µL PBS) or PBS as a control. In the pre-treatment groups, olaparib (1, 3, or 10 mg/kg) or rucaparib (1, 3, or 10 mg/kg) were administered intraperitoneally (i.p.) 1 h before the LPS challenge. In the post-treatment group, olaparib (10 mg/kg, or its control vehicle) was administered intraperitoneally 1, 2, or 3 h after the LPS challenge. All the animals were sacrificed 24 h after LPS instillation. Lungs were collected for histological analysis, morphometry, cytokine quantification, and Western blotting analysis. The experimental design is graphically represented in Figure 1.

Each group contained approximately 50% male and 50% female animals and was considered together as a single group. In a post hoc analysis, male and female groups were also analyzed separately. Each animal group contained 8–12 animals. In total, 240 animals were used for the current project.

### 2.5. Bronchoalveolar Lavage Fluid (BALF) Collection and Analysis

After 24 h, the animals received an intraperitoneal dose of pentobarbital (150 mg/kg) and bled by transection of the inferior vena cava. Bronchoalveolar lavage was performed by the i.t. instillation of 1 mL PBS into the lungs. The lavage fluid was infused a total of two times into the lungs before the final collection. BALF was then centrifuged at 5000 rpm for 10 min, and the cell-free supernatant was frozen at −80 °C until further analysis. The volume of cell-free supernatant was measured for each animal.

### 2.6. Histology

After BALF collection, the trachea was cannulated, and the lungs were fixed by the instillation of 0.5 mL of buffered formalin (10%), at a pressure of 18–22 cm H_2_O, for 2 min. The trachea was then ligated, and the lungs, separated from the heart, were immersed in buffered formalin (10%) solution for 48 h. The left lung was embedded in paraffin, sliced (5 µm) perpendicularly to the lung base (apico–basal axis), giving origin to 2 halves with portions of the upper, middle, and base of the lung, and stained with hematoxylin–eosin (H&E).

### 2.7. Histomorphometry

To access uniform and proportional lung samples, 10 fields (five non-overlapping fields in two different sections) were randomly analyzed in the proximal and distal lung parenchyma (airways and AECs). The measurements were conducted with a 100-point and 50 straight grid with a known area (62.500 mm^2^ at a 400× magnification) attached to the ocular of the microscope. At 400× magnification, the area in each field was calculated according to the number of points hitting positive cells for specific antibody as a proportion of the total grid area. Bronchi and blood vessels were carefully avoided during the measurements. In order to normalize the data, the area occupied by each specific antibody, measured in each field, was divided by the length of each septum studied (to avoid any bias secondary to septal edema, alveolar collapse and denser tissue matrix seen in the fibrotic sections). The results express the fractional area of positive cells determined as the number of positive cells in each field divided by the connective tissue area. The results were expressed as percentages.

### 2.8. Quantification of DNA Damage In Vivo

For in situ detection of DNA fragmentation at the level of a single cell, the TUNEL method was performed using the ApopTag^®^ peroxidase kit (Intergen Co., Oxford, United Kingdom), following the manufacturer’s instructions. Briefly, thick paraffin sections (5 µm) were layered on glass slides, deparaffinized with xylene, and rehydrated with graded dilutions of ethanol in water. Protein digestion was conducted by incubating tissue sections in 20 μg/mL proteinase K (Promega, Madison, WI, USA) for 15 min at room temperature, and endogenous peroxidase was inactivated with 2% H_2_O_2_ in distilled water (dH_2_O) for 5 min, at room temperature. The labeling mixture containing biotinylated dUTP in TdT enzyme buffer was added to sections and incubated at 37 °C in a humified chamber for 1 h, following the manufacturer’s instructions. The sections were then incubated with anti-digoxigenin peroxidase conjugate and incubated for 30 min at room temperature in a humified chamber. The sections were then incubated in TBS with 0.05% diaminobenzidine (DAB) + 3% H_2_O_2_ until color development was achieved. The reaction was terminated by washing the sections twice in phosphate-buffered saline, counterstained in hematoxylin, dehydrated, and mounted with DPX (Panreac SA, Barcelona, Spain). The staining was visualized with a substrate system in which nuclei with DNA fragmentation stained brown. The nuclei without DNA fragmentation stained blue as a result of counterstaining with hematoxylin. For negative control, TdT was eliminated from the reaction mixture and replaced by the kit equilibration buffer.

### 2.9. Quantification of Pulmonary Extravasation

The concentration of proteins in the BALF, determined using the Bradford assay, was expressed as micrograms of protein per milliliter of BALF.

### 2.10. Enzyme-Linked Immunosorbent Assay (ELISA)

The concentration of the pro-inflammatory cytokines IL-1, IL-6, and TNF-α in the BALF was determined using commercially available ELISA, according to the protocol provided by the manufacturer (R & D Systems, Minneapolis, MI, USA).

### 2.11. Western Blotting

The total protein content in the lung tissue extracts was determined by the Bradford assay (Bio-Rad Protein Assay Dye Reagent Concentrate). The lysates were denatured in lithium dodecyl sulfate (LDS) sample buffer (Bolt™ LDS Sample Buffer, Invitrogen, Thermo Fisher Scientific, Basel, Switzerland) or sodium dodecyl sulfate (SDS) sample buffer (Novex™ Tris-Glycine SDS Sample Buffer, Invitrogen, Thermo Fisher Scientific, Basel, Switzerland) supplemented with 50 mM dithiothreitol (DTT; Bolt™ Reducing Agent, Invitrogen, Thermo Fisher Scientific) at 95 °C for 5 min. Samples (20 μg of total protein from whole extracts) were resolved in SDS-PAGE, and the proteins were transferred to nitrocellulose or PDVF membranes by dry transfer using the iBlot™ 2 Device and Transfer Stacks nitrocellulose or PVDF (Invitrogen, Thermo Fisher Scientific). Membranes were blocked with Tween TBS containing 5% milk and probed with the specific primary antibodies overnight at 4 °C (NF-κB p65 1:1000, NLPR3 1:1000, caspase 3 1:1000, Akt-1 1:1000, pAKT-1 (Ser473-Akt1 specific) 1:1000, β-catenin 1:1000). The source of all antibodies was Cell Signaling Technology (Danvers, MA, USA). Catalog numbers for the antibodies were the following: NF-κB p65: D14E12 rabbit—catalog number: 8242; NLPR3: D4D8T rabbit—catalog number: 15101; caspase 3: rabbit—catalog number: 9662; Akt-1: 73H10 rabbit—catalog number: 2938; pAKT-1 (Ser473-Akt1 specific): D7F10 rabbit—catalog number: 9018; β-catenin: D108 rabbit—catalog number: 8480. After extensive washing in Tween TBS, membranes were incubated with anti-rabbit IgG or anti-mouse IgG, HRP-linked antibody diluted 1:5000 in 5% milk-TBST and incubated 1 h at room temperature. Immunoreactive proteins were visualized by Radiance Plus (AC2103, Azure Biosystems, Dublin, CA, USA) chemiluminescence solution using an Azure Imaging System 300 (Azure Biosystems). Specific bands were quantified by densitometry using ImageJ (NIH, Bethesda, MD, USA).

### 2.12. Statistical Analysis

Data shown are mean ± SEM and are representative of the number of cell cultures or animals used per experimental group. Differences were analyzed using ANOVA followed by post hoc analysis using Tukey’s *t*-test. Statistical significance was assigned to *p* < 0.05.

## 3. Results

### 3.1. Cytoprotective Effects of PARP Inhibitors In Vitro

In U937 cells, none of the 5 clinically used PARP inhibitors tested (1–30 µM, 2 h, or 24 h) affected the viability of the cells under baseline conditions (i.e., in the absence of oxidative stress) (Figure 2). In these cells, 600 µM H_2_O_2_ produced an approximately 50–80% decrease in cellular MTT converting activity at 2 and 24 h, respectively. Under oxidative stress conditions (H_2_O_2_ challenge for 2 h or 24 h), the decrease in MTT conversion and the increase in LDH content in the supernatant (indicative at cell necrosis at 24, but not at 2 h) was counteracted by all PARP inhibitors tested. The effects on MTT conversion were more pronounced at 2 h than at 24 h (Figure 2). Most of the efficacy of the inhibitors was already achieved at the lowest concentration of the inhibitor tested (1 µM); further increases in the concentration of the inhibitor did not yield additional protection. There were no major differences in the degree of protective efficacy between the 5 PARP inhibitors studied, although the most pronounced protection was achieved with olaparib at the lowest concentration used (1 µM) (Figure 2, Figure 3, Figure 4 and Figure 5).

In the BEAS-2B cells (a human pulmonary epithelial cell line), similarly to the U937 cells, none of the 5 PARP inhibitors tested (1–30 µM, 2 h, or 24 h) significantly affected the viability or MTT-converting activity under baseline conditions (i.e., in the absence of oxidative stress), although some of the inhibitors, at the highest concentration used, tended to decrease cell viability (Figure 6, Figure 7, Figure 8 and Figure 9).

This cell line was more sensitive to oxidative stress than the U937 cells; already, 30 µM H_2_O_2_ produced an approximately 50–80% decrease in MTT-converting activity at 2 and 24 h, respectively.

In this cell line—in contrast to the findings in U937 cells—only a partial cytoprotective effect of PARP inhibition was observed for some of the other PARP inhibitors tested (rucaparib, talazoparib); two of the inhibitors (olaparib and veliparib) only exerted a slight protective effect at the lowest, or the highest concentration tested, respectively, and only on the MTT assay and not the LDH assay, while one PARP inhibitor (niraparib) was without significant protective effect in this cell line (Figure 6, Figure 7, Figure 8 and Figure 9).

Because of the well-known role of PARP in the recruitment of DNA repair enzymes and in the maintenance of genetic integrity, the effect of two selected PARP inhibitors, olaparib and rucaparib, was also tested on DNA integrity. DNA damage in response to the H_2_O_2_ challenge (600 µM in U937 cells and 30 µM in BEAS-2B cells) was comparable in both cell types studied and induced approximately 2–3-fold increases in DNA strand breakage over baseline levels.

The PARP inhibitors, at the concentrations where they exerted cytoprotective effects in the MTT and LDH assays (3 and 30 µM), did not potentiate the degree of H_2_O_2_-induced DNA damage, but, in fact, unexpectedly, reduced the degree of DNA damage both in U937 cells (Figure 10A) and in BEAS-2B cells (Figure 10B), although in some instances clear concentration-dependence of the response was not observed.

Importantly, none of the PARP inhibitors used increased/potentiated the DNA damage induced by the oxidant.

### 3.2. Cytoprotective Effects of PARP Inhibitors In Vivo

We have selected two of the clinical-stage PARP inhibitors, olaparib, and rucaparib, for subsequent in vivo evaluation in a murine ALI model. Pre-treatment with either of these two PARP inhibitors improved the histological status of the lungs during the LPS challenge by reducing the number of infiltrating inflammatory cells (Figure 11).

The PARP inhibitors also attenuated pulmonary extravasation (protein content measured in the bronchoalveolar lavage fluid); the effects were statistically significant—albeit only partial—at the highest dose of either of the two inhibitors tested (Figure 12).

The PARP inhibitors also reduced the number of neutrophils—but not mononuclear cells—in the bronchoalveolar lavage fluid. These effects were already noted at lower doses of the PARP inhibitors used; olaparib tended to be more effective than rucaparib in this regard (Figure 12).

Quantification of DNA damage in the lungs shows that the PARP inhibitors did not increase the degree of DNA damage but, rather, decreased it in a concentration-dependent manner, once again, with olaparib being more effective than rucaparib (Figure 13 and Figure 14).

Both PARP inhibitors reduced TNF-α, IL-1β and MIP-1α levels in the bronchoalveolar lavage fluid, while IL-6 levels were largely unaffected; overall, the efficacy of olaparib was higher than that of rucaparib (Figure 15).

The anti-inflammatory and morphological benefit of the PARP inhibitors was linked to complete or near-complete inhibition of NF-κB p65 expression as well as caspase 3 expression, suggesting that PARP inhibitors exert a suppressive effect on the activation of these pathways (Figure 16). However, the MAP kinase and Akt pathways were not significantly affected by PARP inhibitors (Figure 16).

NLRP3 induction was significant after the LPS challenge, and this effect was completely abolished by both PARP inhibitors—at 3 and 10 mg/kg olaparib and at 1, 3, and 10 mg/kg rucaparib (Figure 17). In addition, the LPS-induced increase in the expression of β-catenin was also completely inhibited by both of the PARP inhibitors used in both concentrations tested (Figure 17).

We next assessed if the efficacy of the PARP inhibitor is maintained when its administration is delayed relative to the start of the ALI induction by LPS. Because olaparib was more effective on several parameters than rucaparib, in these experiments, olaparib was used; the dose was selected as the highest dose tested in the previous experimental runs (10 mg/kg).

Delaying the administration of olaparib to 1, 2, or 3 h post-LPS remained effective in terms of reducing histological damage to the lung (Figure 18).

Olaparib also remained effective in terms of reducing pulmonary extravasation and inhibiting neutrophilia in the BAL (Figure 19).

Moreover, olaparib post-treatment—similar to its effect in pre-treatment shown above—did not exacerbate but, rather, attenuated the degree of DNA injury in the pulmonary tissue (Figure 20).

Inhibition of IL-β levels in the BAL (but not the inhibition of TNFα) was maintained in post-treatment (Figure 20). PARP inhibition also suppressed IL-6 and MIP-1α production in the various post-treatment protocols (1–3 h after LPS) (Figure 21).

The inhibition of NF-κB p65 expression and caspase 3 expression were also maintained in the post-treatment paradigm (Figure 22). Similarly to the pre-treatment paradigm, the ERK and Akt pathways were also not inhibited by olaparib in the post-treatment regimen (Figure 22); in fact, ERK1/ERK2 ratio gradually increased by olaparib in the various post-treatment paradigms with the most pronounced effect noted when the inhibitor was applied 3 h post-LPS (Figure 22).

NLRP3 induction also remained suppressed by olaparib in the post-treatment paradigm (Figure 23), and so was β-catenin expression (Figure 23). Taken together, olaparib’s efficacy is generally maintained when its administration is delayed by 1, 2, or 3 h relative to the initiation of ALI.

We utilized combined groups of male and female animals. In a number of preclinical models, PARP inhibitors’ protective effect shows a significant gender difference (efficacy being more pronounced in male than female animals) [14,15,16]. Therefore, we have conducted a post hoc sub-group analysis for some of the parameters that were significantly affected by olaparib in the current ALI model. There were some sex differences in the effect of olaparib or rucaparib in the pre-treatment paradigm; in male animals, the inhibitory effect of the PARP inhibitors on extravasation and on TNFα production tended to be more pronounced than in female animals (Figure 24). With regard to the other measured analytes, no sex differences were noted with respect to the effect of olaparib (data not shown).

No significant sex differences were seen in olaparib’s effect in the post-treatment paradigm (Figure 25).

## 4. Discussion

The principal conclusion of the current study is that clinically approved PARP inhibitors, such as olaparib and rucaparib, exert protective effects in various cell types against oxidative stress in vitro and protect the lung in an experimental model of LPS-induced lung injury. These data confirm and extend prior studies demonstrating the protective effect of various earlier-generation PARP inhibitors, or PARP1 deficiency [10,14,15,16,17,18,19,20,21,22,23,24,25,26,27], or clinically approved PARP inhibitors such as olaparib [28,29,30,31,32] in various models of lung injury (reviewed in [10]). Although many prior studies have utilized various PARP inhibitors in various models of lung injury (and have already demonstrated that such agents protect against extravasation, pro-inflammatory pathway activation, infiltration of various immune cells [10]), specifically the available information regarding clinically approved PARP inhibitors in ALI (or, more generally, in models of lung injury) is rather limited and consists of three studies: (a) a rat model of LPS-induced lung injury, where—similarly to the current study—olaparib was found to decrease the activation/production of several pro-inflammatory signalling pathways/genes including NF-κB and TNF-α [28]; (b) a mouse model of ovalbumin-induced chronic asthma, where olaparib—similar to the current study—reduced the pulmonary infiltration of various immune cells and suppressed the activation of the NF-κB pathway and of the activation of the NLRP3 system [29] (c) a mouse model of sepsis induced by cecal ligation and puncture, where olaparib—similar to the current study—improved lung histology and exerted systemic beneficial effects without exacerbating DNA injury, as assessed by the TUNEL assay [30]; (d) a mouse model of ALI induced by intratracheal LPS administration, where—similarly to the current study—olaparib was found to decrease the activation/production of several pro-inflammatory signalling pathways/genes including NF-κB and TNF-α (and in addition, PARP inhibition also suppressed various oxidative stress markers and improved central nervous system function) [31] and (e) a model of influenza virus-induced pneumonia, where olaparib—similarly to the current study—olaparib was found to decrease the activation/production of several pro-inflammatory signalling pathways/genes including NF-κB and TNF-α [32]. The current study confirms and extends many of the above findings. Importantly, however, none of the prior studies compared the effect of various clinically used PARP inhibitors. Moreover, the prior studies have focused on the efficacy aspect of PARP inhibition and did not examine safety-related issues (e.g., the issue of DNA damage and its potential potentiation by PARP inhibition). Moreover, prior studies have not examined the therapeutic window of administration—i.e., the administration of the PARP inhibitor started early on, and its efficacy was not evaluated when its administration was delayed to later time points relative to the initiation of the injury. Finally, prior studies focusing on the effect of olaparib on various forms of lung injury typically only used one gender of animals (males) and did not investigate sex differences in the effect of the PARP inhibitor.

Although all 5 PARP inhibitors used in the current study have comparable nanomolar potency on the isolated PARP1 enzyme, they also have significant differences with respect to their effects on other PARP family members (e.g., PARP2 and other minor PARylating enzymes). In addition, these inhibitors exhibit important differences in terms of their ability to induce PARP inhibition vs. PARP ‘trapping’ (a phenomenon whereby the PARP enzyme-inhibitor complex “locks” onto damaged DNA and prevents DNA repair, replication, and transcription, potentially leading to cell death) [33,34,35]. From a theoretical, mechanistic standpoint, inhibition of PARP—which serves as an integral part of the DNA repair process, involved, for instance, in the recruitment of various DNA repair enzymes to the site of DNA injury—is expected to induce or potentiate DNA damage and such adverse effects on DNA integrity are expected to be more pronounced with PARP inhibitors that have a higher PARP-trapping capacity [4,36,37]. In contrast to these expectations, surprisingly, our results do not demonstrate any potentiation of DNA breakage in oxidatively stressed cells in vitro or in the lung tissue subjected to LPS-induced ALI ex vivo. In fact, the degree of DNA damage was lower, rather than higher, after PARP inhibition, and there were no marked differences between the various PARP inhibitors tested with respect to this response.

The mechanism by which PARP inhibitors protect cells or tissues from oxidative damage is complex. In vitro, in simple oxidant-induced cell injury models, the simplest explanation for the cytoprotective effects of PARP inhibitors follows the “Berger hypothesis”, whereby PARP inhibitors prevent the NAD^+^ and ATP depletion that is elicited by PARP over-activation when PARP recognizes multiple DNA strand breaks [1,2,5,38,39,40,41,42,43], while in vivo, additional effects (e.g., inhibition of pro-inflammatory mediator production, inhibition of immune cell infiltration into the affected tissues, interruption of positive feedforward cycles of inflammation and organ injury) may also contribute [1,5,44,45,46,47,48]. Under such conditions, we hypothesize that PARP inhibition maintains overall improved cell viability and may also maintain a better cellular bioenergetic profile, which, in turn, may help maintain the activity of various DNA repair enzymes.

The effect of the PARP inhibitors was not uniform on all of the parameters measured. For instance, some of the inflammatory mediators were affected more than others. This may be related to the fact that the signaling pathways involved in the generation of these mediators are different, and PARP activity may have a differential role in modulating these pathways. Cell-type-dependent differences were also noted in the ALI model: the infiltration of neutrophils was significantly affected, but mononuclear cells were not uniformly affected by the PARP inhibitors. We do not know the exact reason for this difference, and this remains to be further studied. One potential working hypothesis is that in the current experimental model, PARP plays a more significant role in the regulation of those factors (e.g., soluble mediators and/or cell surface adhesion molecules) that are relevant for neutrophil trafficking, as opposed to those factors that regulate mononuclear cell trafficking. One difference may be important: mononuclear cells have PARP1 expressed in their nucleus, while neutrophils lack this enzyme [1,2]. Thus, the function and trafficking of neutrophils may be mainly influenced by the effects of the PARP inhibitors on other cell types, while the function and trafficking of mononuclear cells may be affected by PARP through a combination of effects on these cells (as well as on their environment). Differential effects of olaparib on neutrophils vs. mononuclear cells have also recently been noted in a model of colitis: in this model, neutrophilia was not affected by the PARP inhibitor, while the changes in lymphocyte and monocyte numbers were partially normalized by the PARP inhibitor [43]. In another study investigating various white blood cell types from myelodysplastic patients, olaparib more significantly affected cells of the myeloid than of the lymphoid lineage [49]. Thus, clearly, olaparib exerts effects on various immune cells in a cell-type- and context-dependent manner. Although the mechanism by which PARP inhibition differentially affects neutrophils vs. monocytes (inhibits their recruitment or does not affect their recruitment, respectively) remains to be investigated, this differential effect may be, in fact, beneficial with respect to the modulation of the inflammatory process in ALI. Considering that in this model of ALI induced by LPS (i.t. and 24 h), the main inflammatory cell recruited are neutrophils, and they play a fundamental role in the installation of the inflammatory process. After this initial peak, the most important role of macrophages is pathogen clearance. Furthermore, macrophage phagocytosis of neutrophils and other apoptotic cells is an important step in the process of inflammatory response regulation [50,51]. Thus, inhibition of neutrophil but not macrophage infiltration, in fact, may be a preferred modulation of the ALI response than, for instance, an indiscriminate inhibition of the recruitment of all immune cells.

Interestingly, the degree of protection afforded by PARP inhibitors against oxidative stress was different in the two cell lines investigated; in the U937 cells, the effects were more pronounced than in the BEAS-2B cells—consistent with prior data demonstrating that the role of PARP activation in mediating cell death (and, indeed, various forms of cell death from cell necrosis to parthanatos) can be different in different cell types and tissues [1,2,3,4,5]. Nevertheless, we found that the protection against oxidative stress-induced DNA damage is significant in both cell types studied, although it is not uniformly present with all of the clinically used PARP inhibitors evaluated. Interestingly, in a recent independent study by Kondratska and colleagues, utilizing a systemic administration of LPS in mice, granulosa cell DNA damage, once again, was not exacerbated but rather inhibited by the PARP inhibitor 4-hydroxyquinazone [52]. We are aware that only one method of DNA damage was used in the current study—i.e., the TUNEL assay, which is generally not considered of particularly high sensitivity to assess DNA damage [53,54,55]. Nevertheless, the assay was sensitive enough to detect the DNA damage induced by injury (oxidative stress, ALI), and directionally the effect of PARP inhibition was to decrease, rather than increase, the degree of DNA breakage. The above considerations notwithstanding, we must point out that other parameters related to DNA damage or chromosomal integrity (e.g., DNA base modifications, chromosomal stability, micronucleus formation, and others) have not been assessed in the current project; these additional parameters would be important to examine prior considering clinical translation of PARP inhibitors for the experimental therapy of non-oncological diseases.

As far as the therapeutic window of PARP inhibitors’ administration, the current study reveals that delaying the start of PARP inhibition to 3 h post-LPS in the current model of ALI essentially produced identical protective effects when compared to earlier administration of the inhibitor. These data are translationally relevant because they suggest that the start of a PARP inhibitor therapy in a clinical setting (in the paradigm of therapeutic repurposing) can be delayed to later time points at which the diagnosis of ALI is confirmed, and the inhibitor may provide benefit even with delayed administration. Further studies remain to be conducted to determine if further delay of its administration (e.g., to 6 or 8 h after the initiation of ALI) may also provide therapeutic benefit in the current model.

The efficacy of PARP inhibitors in the post-treatment setting also indicates that the site of PARP inhibitors’ beneficial effect in the current model includes cellular and molecular targets that are continuously ongoing and/or become later activated during the sequelae of various pathophysiological events, as opposed to interfering with early triggers of injury. The pathophysiological pathways that were affected by PARP inhibition both in co-treatment and post-treatment regimens included many pathways and targets which have previously been implicated in the anti-inflammatory or therapeutic effect of PARP inhibitors in various non-oncological disease models, including NF-κB [24,44,45,56,57]; the production of various chemokines, such as MIP-1α [46]; modulation of the production of various cytokines [14,15,16,17,29,30,47]; inhibition of inflammatory cell mobilization and recruitment [7,8,14,29,58]; and more recently implicated pathways, including the activation of the NLRP3 inflammasome [29,59]. Regarding the effect of PARP inhibition on the expression of β-catenin, the current study shows that PARP inhibition suppresses the expression of this mediator during ALI. (It should also be mentioned that previous studies have already observed an inhibitory effect of olaparib on β-catenin expression in various cancer cells [60,61], and in this context, the effect of the PARP inhibitor was implicated in the process of epithelial-to-mesenchymal transition, an important component of cancer cell metastasis.) The wnt/β-catenin pathway has been implicated in the pathogenesis of ALI and sepsis as a pro-inflammatory mechanism that contributes to the development of injury [62,63,64], although the same pathway is also known—at later stages of the lung injury—to play a beneficial role in lung repair and regeneration, in part through the regulation of epithelial/mesenchymal transition [65,66]. Whether the net effect of olaparib’s inhibitory effect on β-catenin expression in various stages of ALI remains to be further investigated.

Although PARP has also been implicated in the stimulation of Akt activation, and via this pathway, it may confer mitochondrial stabilization and cytoprotection [17,19,67,68,69], in the current experimental system, olaparib did not have any marked effect in the pre-treatment protocol and induced a slight activation of the pathway in post-treatment. It is possible that Akt activation may be a component of the protective effect of the PARP inhibitor in the delayed (but not in the earlier) treatment paradigms.

Another translationally relevant issue is the question of potential gender differences in the effect of PARP inhibitors in non-oncological situations. The first studies to recognize this issue have been conducted in murine models of stroke, where PARP inhibition or PARP1 deficiency produced marked benefit in male animals, but, paradoxically, in female animals, the protection was absent, and, in fact, in some cases, exacerbation of the injury was noted [47,70,71,72,73]. In a model of LPS-induced systemic inflammation, male mice benefited from the PARP inhibitor PJ34 more than female mice (unless female mice were subjected to ovarectomy, in which case the LPS-induced injury became more severe, and the protective effect of the PARP inhibitor was restored) [47]. The mechanism of this sex difference was attributed, at least in part, to the modulatory role of estrogen in the ability of PARP to recognize DNA strand breaks [47]. However, the sex difference in the efficacy of PARP inhibitors was not general, as in several studies, PARP inhibitors were also found to be efficacious in female animals [74]—including in an ovine model of ARDS [18,23]. Indeed, in the current study, for most parameters—with the exception of TNF1α, which tended to be more suppressed by PARP inhibitors in males than in females—no marked differences were noted in the degree of injury when the male and female sub-cohorts were compared, and the PARP inhibitors tested have demonstrated efficacy in both groups.

Some of the strengths of the current study include the facts that (1) all clinically used PARP inhibitors were tested in vitro, (2) the in vivo model incorporated a translationally relevant post-treatment paradigm, and (3) that the in vivo model included animals of both sexes, and potential sex differences in the responses were included in the experimental design. Some of the limitations of the current study include the fact that (1) in vivo, not all of the clinically used PARP inhibitors were tested, only two selected agents; (2) the ALI model used in the study utilized LPS, which is only one (albeit important) component of the bacteria involved in the production of inflammatory responses and organ injury; (3) only one DNA damage assay (TUNEL) was used, and other modes of potential DNA damage elicited by PARP inhibition (e.g., chromosomal instability) were not tested; and (4) in vivo pharmacokinetic studies were not included into the study, and therefore the plasma levels associated with the efficacy of the PARP inhibitors remain to be determined in the future. (5) Another set of limitations relates to the fact that we have only measured expression levels and/or ratios of various signaling pathways (NF-κB, ERK—but not phosphorylated ERKs–, NLRP3) and not the actual activity of these pathways or the downstream consequences of these pathways. The reason is that direct activity assays of these types are rather difficult to perform in vivo or in tissue homogenates. Nevertheless, there is ample information that the 65kD subunit of NF-κB functions as a potent transcriptional activator and a target for v-Rel-mediated repression [75], and therefore, if this subunit is lower, it is likely that the NF-κB system’s activity is also likely suppressed. Likewise, although the quantification of phosphorylated ERKs and ERK1/2 ratios is more common than the quantification of ERK1/ERK2 ratios, a direct relationship between the ratio of phosphorylated ERKs and the quantitative expression ratio of ERKs has been previously shown [76]; therefore, the ratio measured here may well give some indication regarding the activity of this pathway. With respect to NLRP3, the expression level of this protein correlates well with the activity of the pathway and the downstream pathways affected by it [77,78], and therefore, it is likely that PARP inhibition suppresses the downstream effectors of this pathway as well (although this possibility remains to be directly examined in the future).

Nevertheless, even with these limitations, the conclusions of the study are clear and support the idea of therapeutic repurposing. We do not suggest that from the clinically used PARP inhibitors, olaparib is the only (or best) agent for repurposing, although in the current model, its efficacy, at least on some parameters, appeared to be superior to the efficacy of rucaparib. Nevertheless, olaparib is the first clinically used PARP inhibitor; arguably, the largest body of clinical information is available with this agent, and also in preclinical studies focusing on various non-oncological conditions, this agent was the one that has been utilized most extensively in preclinical studies focused on PARP inhibitor repurposing [9,10,11,12].

## 5. Conclusions

Taken together, the current results demonstrate the preclinical efficacy of various clinically repurposable PARP inhibitors in a model of ALI. The underlying mechanisms are likely multiple: they may involve direct cytoprotective effects, regulation of various inflammatory mediators, and—as a consequence of these—suppression of inflammatory cell infiltration and attenuation of various self-amplifying cycles of cell and tissue injury. These mechanisms have already been characterized extensively in various in vitro and in vivo models using various classes of PARP inhibitors, including clinically approved ones such as olaparib [1,2,5,6,7,8,9,10,11,12,13,14,15,16,17,18,19,20,21,22,23,24,25,26,27,28,29,30,31,32,38,39,40,41,42,43,44,45,46,47,48].

The findings presented in the current report support the concept [5,9,10] that clinically approved PARP inhibitors may be repurposable for the experimental therapy of various non-oncological diseases, including ALI. Importantly, the repurposing concept is further supported by data presented in the current report showing that these inhibitors do not exacerbate DNA damage after endotoxin challenge. These findings are also in line with other recent data [12,52], demonstrating that olaparib—while providing cytoprotective effects and modulatory effects on the production of cytokines in human mononuclear cells—does not impair the ability of these cells to mount an effective antibacterial immune response.

## Figures and Tables

**Figure 1 cells-11-03789-f001:**
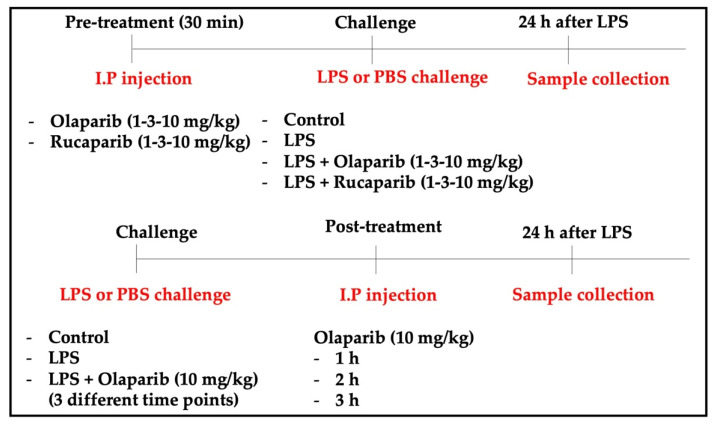
Experimental design used for the in vivo experiments.

**Figure 2 cells-11-03789-f002:**
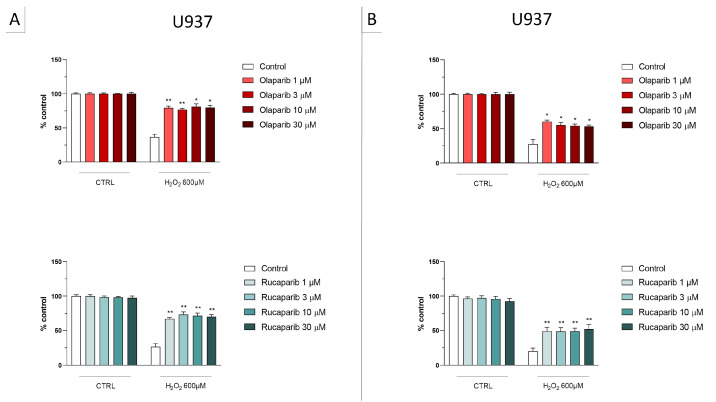
Effect of the clinical-stage PARP inhibitors olaparib or rucaparib on H_2_O_2_-induced suppression of cellular MTT-converting activity in U937 cells. Cells were treated for 2 h (**A**) or 24 h (**B**) with 600 µM H_2_O_2_; the PARP inhibitors were applied at different concentrations (1, 3, 10, and 30 µM). * *p* < 0.05 and ** *p* < 0.01 indicate the beneficial effect of PARP inhibition to counteract the effect of oxidative stress. Data are shown as mean ± SEM of n = 5 independent experiments.

**Figure 3 cells-11-03789-f003:**
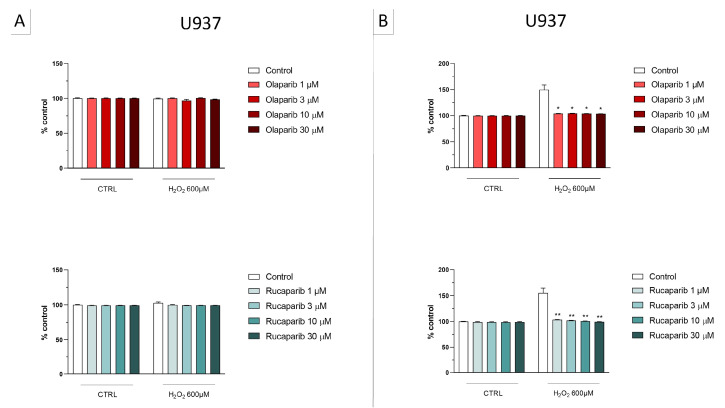
Effect of the clinical-stage PARP inhibitors olaparib or rucaparib on H_2_O_2_-induced cell necrosis in U937 cells. Cells were treated for 2 h (**A**) or 24 h (**B**) with 600 µM H_2_O_2_; the PARP inhibitors were applied at different concentrations (1, 3, 10, and 30 µM). Cell necrosis was measured by measurement of LDH levels in the supernatant. * *p* < 0.05 and ** *p* < 0.01 indicate the beneficial effect of PARP inhibition to counteract the effect of oxidative stress. Data are shown as mean ± SEM of n = 5 independent experiments.

**Figure 4 cells-11-03789-f004:**
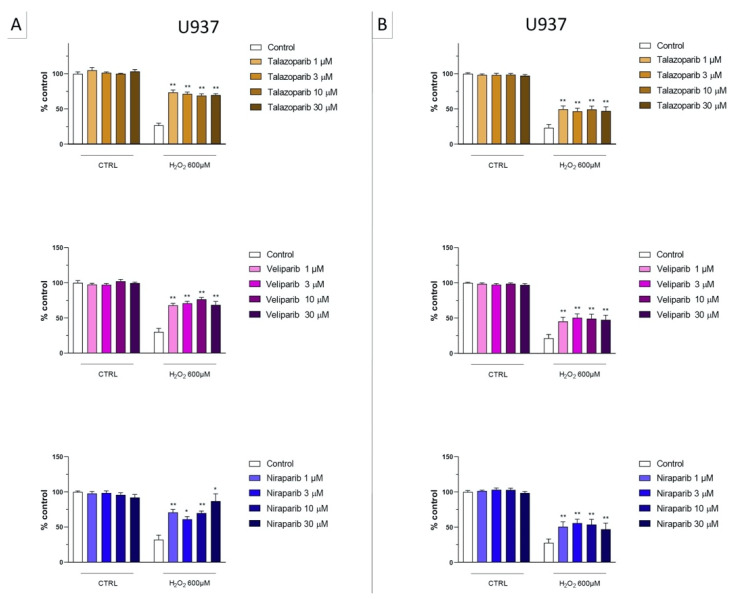
Effect of the clinical-stage PARP inhibitors talazoparib, veliparib, or niraparib on H_2_O_2_-induced suppression of cellular MTT-converting activity in U937 cells. Cells were treated for 2 h (**A**) or 24 h (**B**) with 600 µM H_2_O_2_; the PARP inhibitors were applied at different concentrations (1, 3, 10, and 30 µM). * *p* < 0.05 and ** *p* < 0.01 indicate the beneficial effect of PARP inhibition to counteract the effect of oxidative stress. Data are shown as mean ± SEM of n = 5 independent experiments.

**Figure 5 cells-11-03789-f005:**
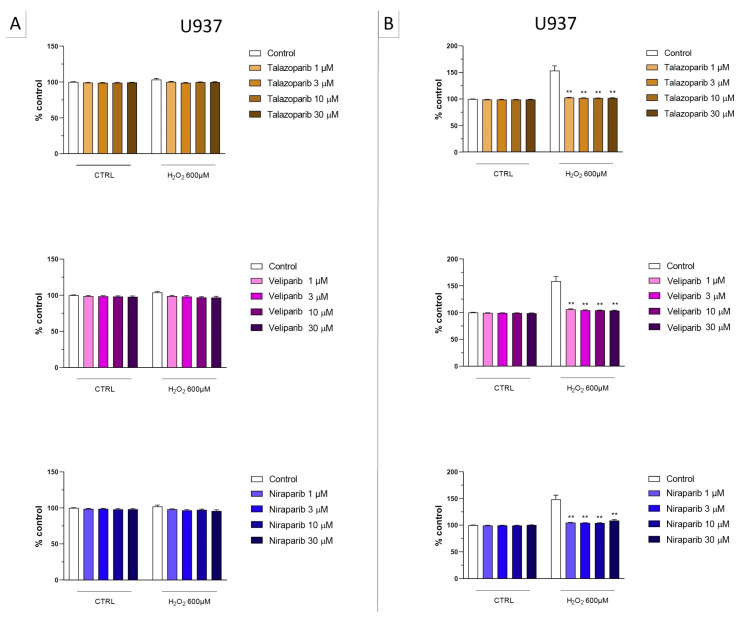
Effect of the clinical-stage PARP inhibitors talazoparib, veliparib, or niraparib on H_2_O_2_-induced cell necrosis in U937 cells. Cells were treated for 2 h (**A**) or 24 h (**B**) with 600 µM H_2_O_2_; the PARP inhibitors were applied at different concentrations (1, 3, 10, and 30 µM). Cell necrosis was measured by measurement of LDH levels in the supernatant. ** *p* < 0.01 indicates the beneficial effect of PARP inhibition to counteract the effect of oxidative stress. Data are shown as mean ± SEM of n = 5 independent experiments.

**Figure 6 cells-11-03789-f006:**
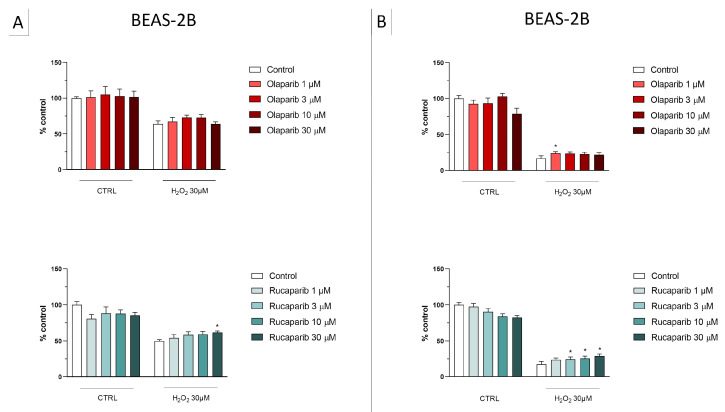
Effect of the clinical-stage PARP inhibitors olaparib or rucaparib on H_2_O_2_-induced suppression of cellular MTT-converting activity in BEAS-2B cells. Cells were treated for 2 h (**A**) or 24 h (**B**) with 30 µM H_2_O_2_; the PARP inhibitors were applied at different concentrations (1, 3, 10, and 30 µM). * *p* < 0.05 indicates the beneficial effect of PARP inhibition to counteract the effect of oxidative stress. Data are shown as mean ± SEM of n = 5 independent experiments.

**Figure 7 cells-11-03789-f007:**
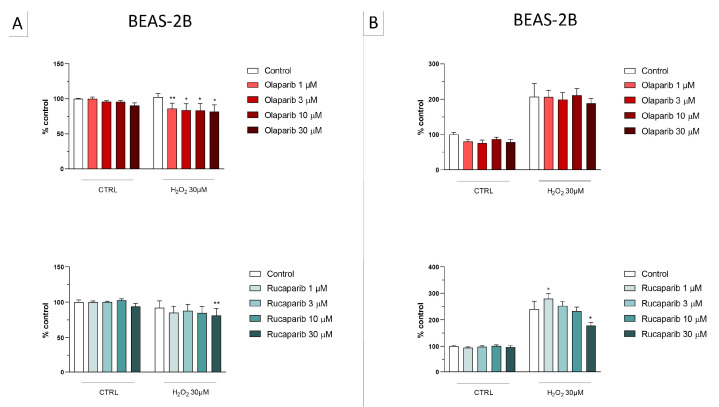
Effect of the clinical-stage PARP inhibitors olaparib or rucaparib on H_2_O_2_-induced cell necrosis in BEAS-2B cells. Cells were treated for 2 h (**A**) or 24 h (**B**) with 30 µM H_2_O_2_; the PARP inhibitors were applied at different concentrations (1, 3, 10, and 30 µM). Cell necrosis was measured by measurement of LDH levels in the supernatant. * *p* < 0.05 and ** *p* < 0.01 indicate the beneficial effect of PARP inhibition to counteract the effect of oxidative stress. Data are shown as mean ± SEM of n = 5 independent experiments.

**Figure 8 cells-11-03789-f008:**
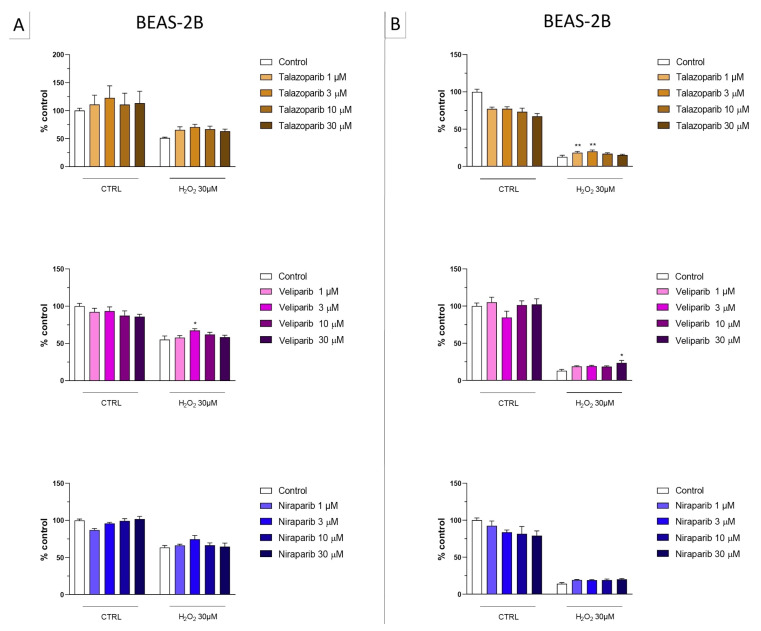
Effect of the clinical-stage PARP inhibitors talazoparib, veliparib, or niraparib on H_2_O_2_-induced suppression of cellular MTT-converting activity in BEAS-2B cells. Cells were treated for 2 h (**A**) or 24 h (**B**) with 30 µM H_2_O_2_; the PARP inhibitors were applied at different concentrations (1, 3, 10, and 30 µM). * *p* < 0.05 and ** *p* < 0.01 indicate the beneficial effect of PARP inhibition to counteract the effect of oxidative stress. Data are shown as mean ± SEM of n = 5 independent experiments.

**Figure 9 cells-11-03789-f009:**
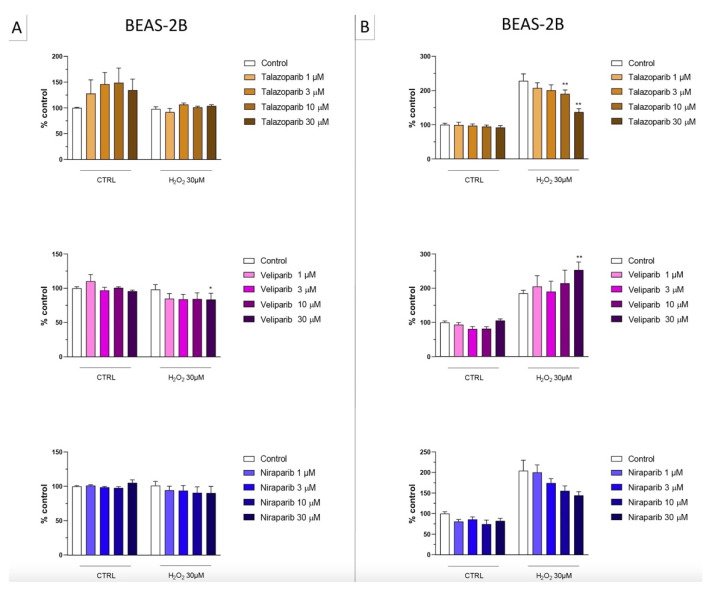
Effect of the clinical-stage PARP inhibitors talazoparib, veliparib, or niraparib on H_2_O_2_-induced cell necrosis in BEAS-2B cells. Cells were treated for 2 h (**A**) or 24 h (**B**) with 30 µM H_2_O_2_; the PARP inhibitors were applied at different concentrations (1, 3, 10, and 30 µM). Cell necrosis was measured by measurement of LDH levels in the supernatant. * *p* < 0.05 and ** *p* < 0.01 indicate the effect of PARP inhibition in oxidatively stressed cells. Data are shown as mean ± SEM of n = 5 independent experiments.

**Figure 10 cells-11-03789-f010:**
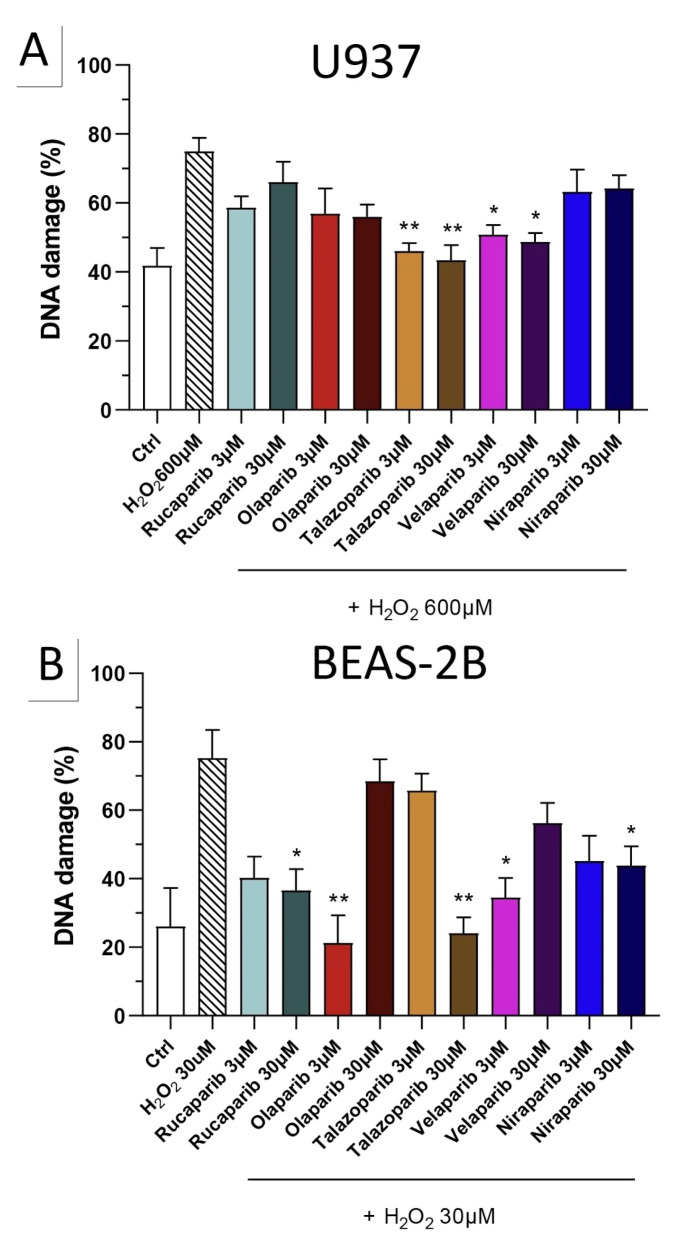
Effect of various clinical-stage PARP inhibitors on H_2_O_2_-induced DNA damage in (**A**) U937 and (**B**) BEAS-2B cells. U937 or BEAS-2B cells were treated for 2 h with 600 µM or 30 µM H_2_O_2_, respectively; the PARP inhibitors were added at different concentrations (3 and 30 µM). After 24 h, DNA damage was measured by the TUNEL assay. * *p* < 0.05 and ** *p* < 0.01 indicate the beneficial effect of the PARP inhibitors tested to reduce the degree of DNA damage in H_2_O_2_-treated cells. Data are shown as mean ± SEM of n = 5 independent experiments.

**Figure 11 cells-11-03789-f011:**
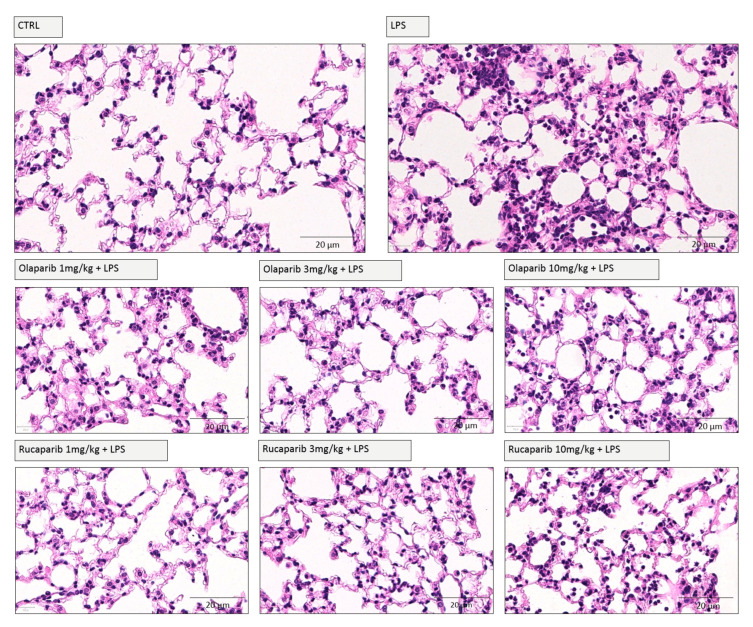
Effect of olaparib and rucaparib on lung histology in a murine model of ALI. C57Bl/6 mice received either PBS (30 µL) or LPS (50 µg in 30 µL PBS) by intratracheal instillation (n = 8 and 12, respectively). After 24 h, lung tissues were analyzed histologically. Representative histological pictures are shown. Please note the increased cell infiltration and inflammation after LPS and the improvement by either of the two PARP inhibitors in the LPS-treated animals.

**Figure 12 cells-11-03789-f012:**
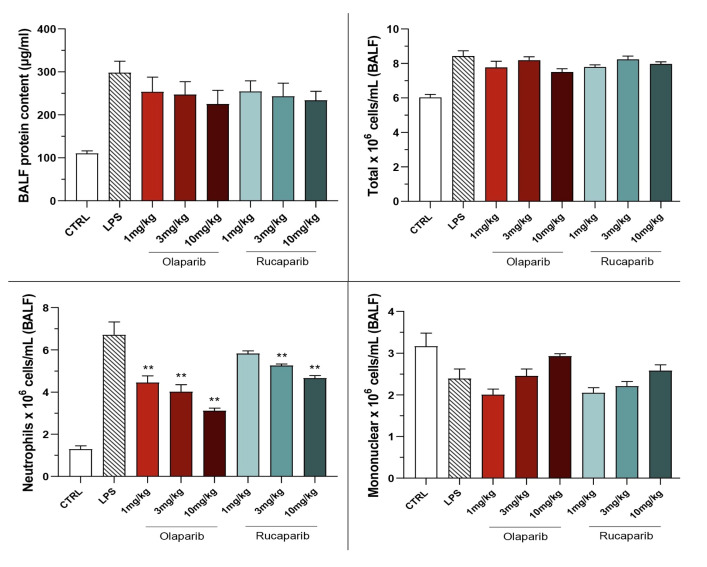
Effect of olaparib and rucaparib on the protein and cell content of bronchoalveolar lavage fluid in a murine model of ALI. C57Bl/6 mice received either PBS (30 µL) or LPS (50 µg in 30 µL PBS) by intratracheal instillation in the absence or presence of various doses (1, 3, or 10 mg/kg) of olaparib or rucaparib. After 24 h, bronchoalveolar lavage fluid was collected, protein content was measured, and inflammatory cell numbers (total cell count as well as neutrophils and mononuclear cells) were quantified. ** *p* < 0.01 indicates the beneficial effect of the PARP inhibitors tested in LPS-treated animals. Data are shown as mean ± SEM, n = 8–12/group.

**Figure 13 cells-11-03789-f013:**
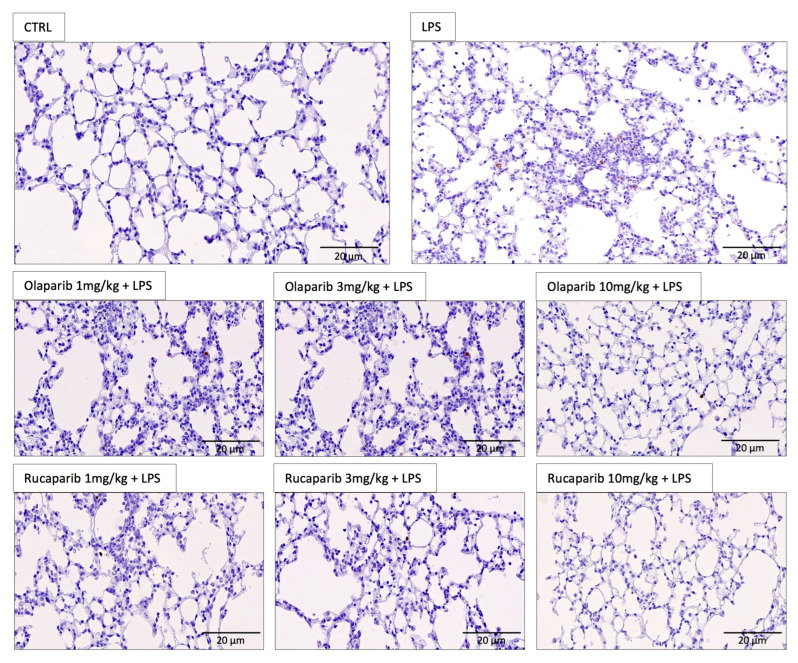
Effect of olaparib and rucaparib on DNA damage in the lung in a murine model of ALI. Representative histological images are shown. Note the increased number of TUNEL-positive cells (brown dots) after LPS and the fewer number of TUNEL-positive cells in the groups treated with the PARP inhibitors. Data were quantified and are expressed in numerical units in Figure 14.

**Figure 14 cells-11-03789-f014:**
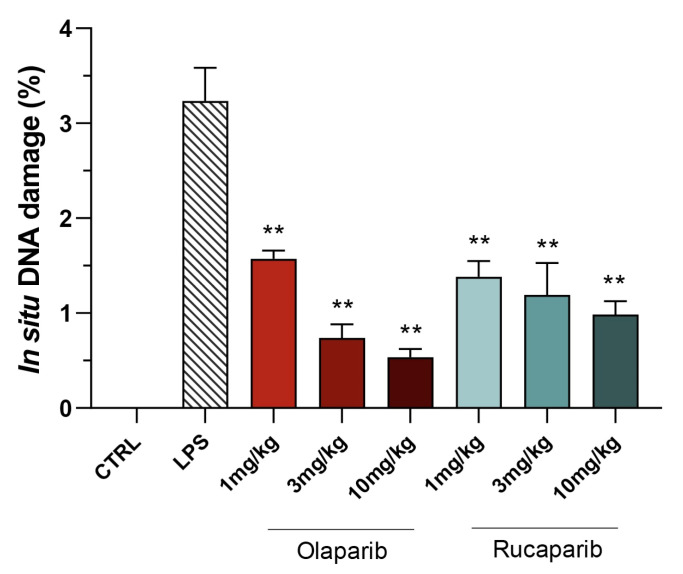
Effect of olaparib and rucaparib on DNA damage in the lung in a murine model of ALI. C57Bl/6 mice received either PBS (30 µL) or LPS (50 µg in 30 µL PBS) by intratracheal instillation in the absence or presence of various doses (1, 3, or 10 mg/kg) of olaparib or rucaparib. After 24 h, DNA damage in the lung was quantified by the TUNEL assay. Top panels show representative histological images. ** *p* < 0.01 indicates the beneficial effect of the PARP inhibitors tested. Data are shown as mean ± SEM, n = 8–12/group.

**Figure 15 cells-11-03789-f015:**
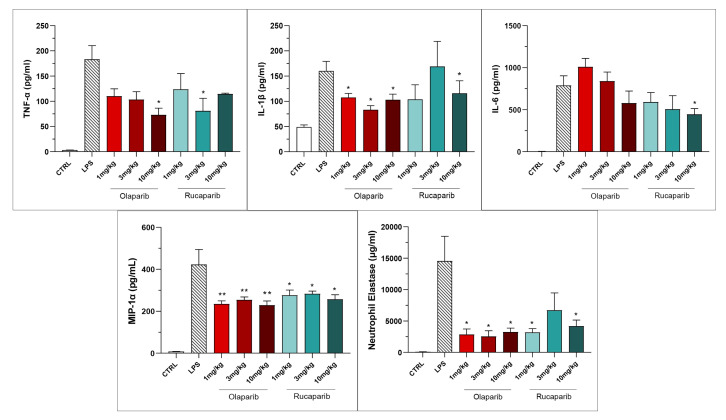
Bronchoalveolar lavage fluid in a murine model of ALI. C57Bl/6 mice received either PBS (30 µL) or LPS (50 µg in 30 µL PBS) by intratracheal instillation in the absence or presence of various doses (1, 3, or 10 mg/kg) of olaparib or rucaparib. After 24 h, bronchoalveolar lavage fluid was collected, and TNFα, IL-1β, IL-6, and MIP-1α or neutrophil elastase levels were measured. * *p* < 0.05 and ** *p* < 0.01 indicates inhibitory effects of the PARP inhibitors tested on various mediator levels in LPS-treated animals. Data are shown as mean ± SEM, n = 8–12/group.

**Figure 16 cells-11-03789-f016:**
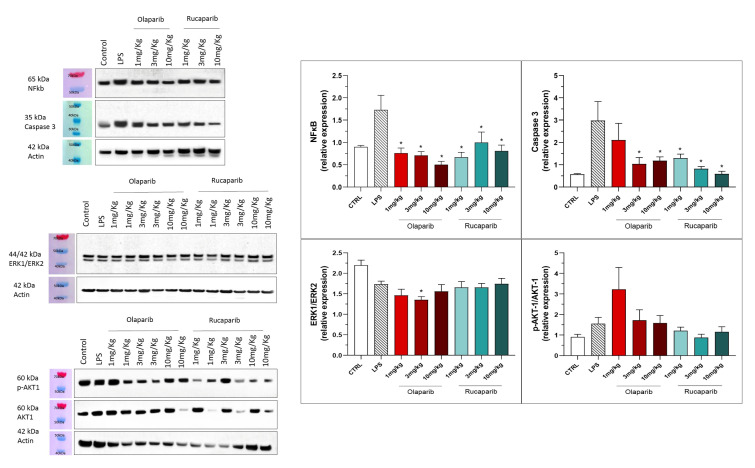
Effect of olaparib and rucaparib on NF-κB, caspase 3, ERK2/ERK1 and pAKT/AKT levels in the lung tissue in a murine model of ALI. C57Bl/6 mice received either PBS (30 µL) or LPS (50 µg in 30 µL PBS) by intratracheal instillation in the absence or presence of various doses (1, 3, or 10 mg/kg) of olaparib or rucaparib. After 24 h, lungs were collected, and NF-κB (65 kDa) and caspase 3 levels, ERK1/ERK2 ratio, and pAKT/AKT ratio were measured by Western blotting. * *p* < 0.05 indicates an inhibitory effect of the PARP inhibitors tested in LPS-treated animals. Data are shown as mean ± SEM, n = 8–12/group.

**Figure 17 cells-11-03789-f017:**
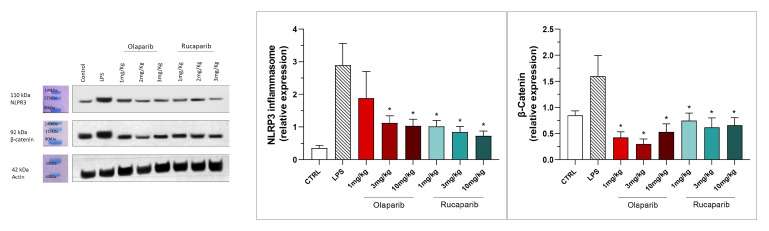
Effect of olaparib and rucaparib on NLPR3 inflammasome activation and β-catenin expression in the lung tissue in a murine model of ALI. C57Bl/6 mice received either PBS (30 µL) or LPS (50 µg in 30 µL PBS) by intratracheal instillation in the absence or presence of various doses (1, 3, or 10 mg/kg) of olaparib or rucaparib. After 24 h, lungs were collected, and NLPR3 inflammasome activation and β-catenin expression were measured by Western blotting. * *p* < 0.05 indicates an inhibitory effect of the PARP inhibitors tested in LPS-treated animals. Data are shown as mean ± SEM, n = 8–12/group.

**Figure 18 cells-11-03789-f018:**
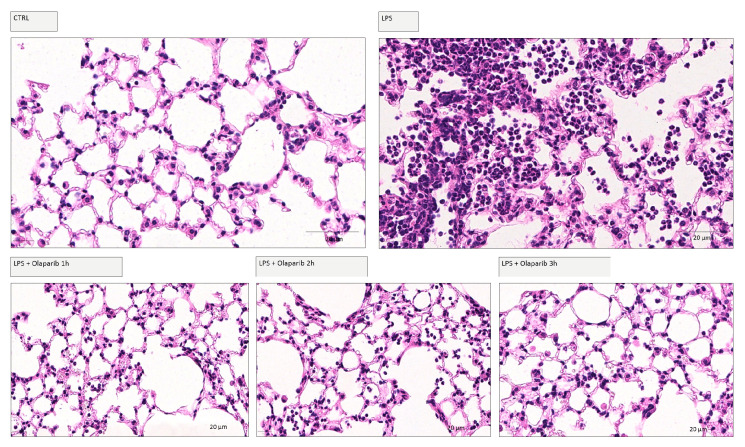
Effect of olaparib post-treatment on lung histology in a murine model of ALI. C57Bl/6 mice received either PBS (30 µL) or LPS (50 µg in 30 µL PBS) by intratracheal instillation in the absence or presence of 10 mg/kg olaparib delayed to 1, 2, or 3 h post-LPS. After 24 h, lung tissues were analyzed histologically. Representative histological pictures are shown. Please note the increased cell infiltration and inflammation after LPS and the improvement by olaparib in the LPS-treated animals.

**Figure 19 cells-11-03789-f019:**
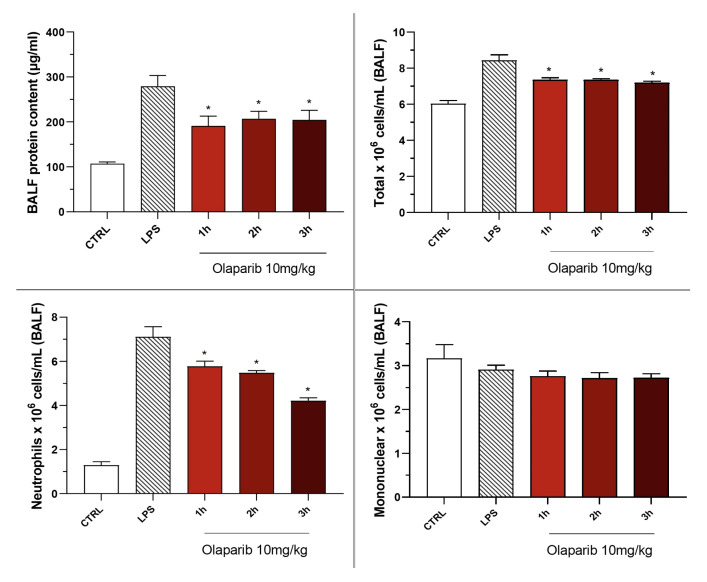
Effect of olaparib post-treatment on the protein and cell content of bronchoalveolar lavage fluid in a murine model of ALI. C57Bl/6 mice received either PBS (30 µL) or LPS (50 µg in 30 µL PBS) by intratracheal instillation in the absence or presence of 10 mg/kg olaparib, delayed to 1, 2, or 3 h post-LPS. After 24 h, bronchoalveolar lavage fluid was collected, and protein content was measured, and inflammatory cell numbers (total cell count as well as neutrophils and mononuclear cells) were quantified. * *p* < 0.05 indicates the beneficial effect of olaparib in LPS-treated animals. Data are shown as mean ± SEM, n = 8–12/group.

**Figure 20 cells-11-03789-f020:**
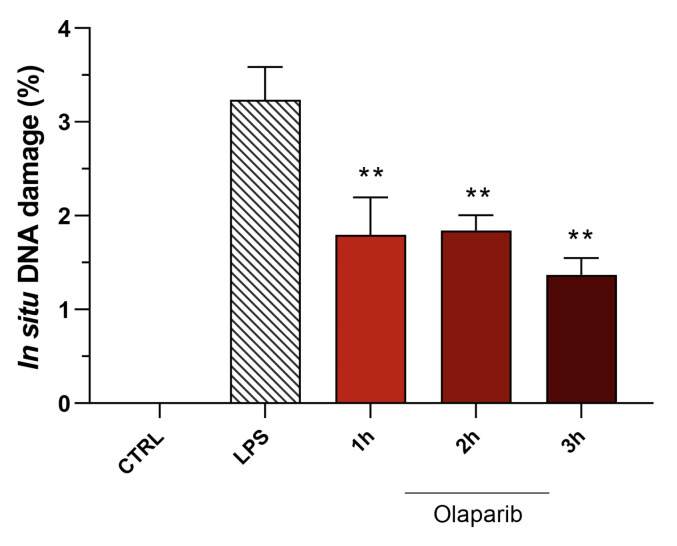
Effect of olaparib post-treatment on DNA damage in the lung in a murine model of ALI. C57Bl/6 mice received either PBS (30 µL) or LPS (50 µg in 30 µL PBS) by intratracheal instillation in the absence or presence of 10 mg/kg olaparib, delayed to 1, 2, or 3 h post-LPS. After 24 h, DNA damage in the lung was quantified by the TUNEL assay. ** *p* < 0.01 indicates the beneficial effect of olaparib against DNA damage in LPS-treated animals. Data are shown as mean ± SEM, n = 8–12/group.

**Figure 21 cells-11-03789-f021:**
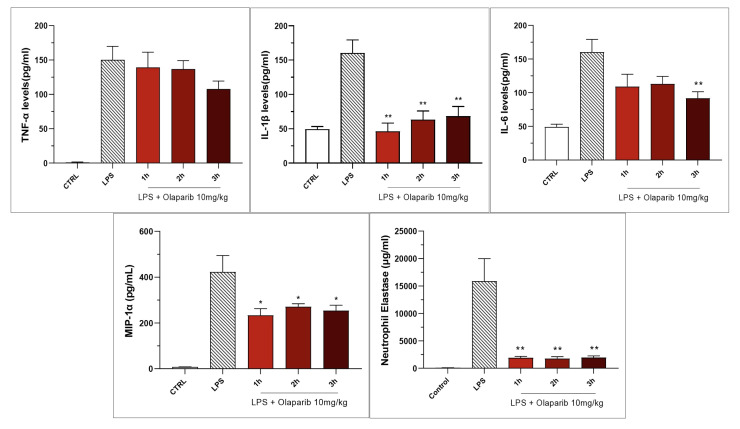
Effect of olaparib post-treatment on selected cytokine and chemokine levels in the bronchoalveolar lavage fluid in a murine model of ALI. C57Bl/6 mice received either PBS (30 µL) or LPS (50 µg in 30 µL PBS) by intratracheal instillation in the absence or presence of 10 mg/kg olaparib, delayed to 1, 2, or 3 h post-LPS. After 24 h, bronchoalveolar lavage fluid was collected, and mediator levels were measured. * *p* < 0.05 and ** *p* < 0.01 indicate the beneficial effect of olaparib in LPS-treated animals. Data are shown as mean ± SEM, n = 8–12/group.

**Figure 22 cells-11-03789-f022:**
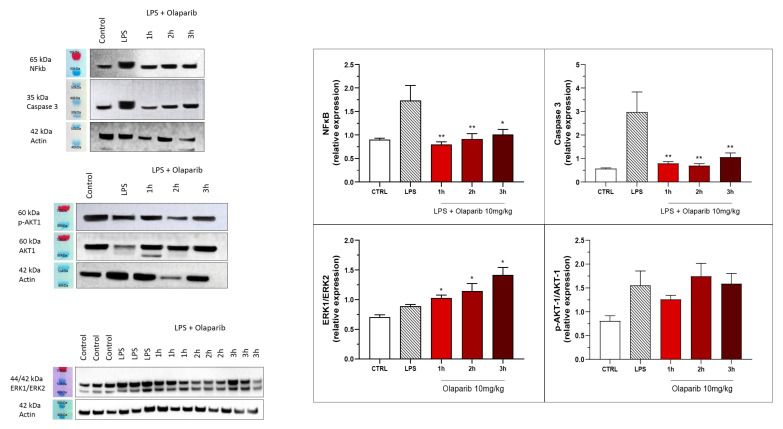
Effect of olaparib post-treatment on NF-κB, caspase 3, ERK2/ERK1, and pAKT/AKT levels in the lung tissue in a murine model of ALI. C57Bl/6 mice received either PBS (30 µL) or LPS (50 µg in 30 µL PBS) by intratracheal instillation in the absence or presence of 10 mg/kg olaparib, delayed to 1, 2, or 3 h post-LPS. After 24 h, lungs were collected, and NF-κB (65 kDa) and caspase 3 levels, ERK1/ERK2 ratio, and pAKT/AKT ratio were measured by Western blotting. * *p* < 0.05 and ** *p* < 0.01 indicate inhibitory effects of olaparib on the various parameters in LPS-treated animals. Data are shown as mean ± SEM, n = 8–12/group.

**Figure 23 cells-11-03789-f023:**
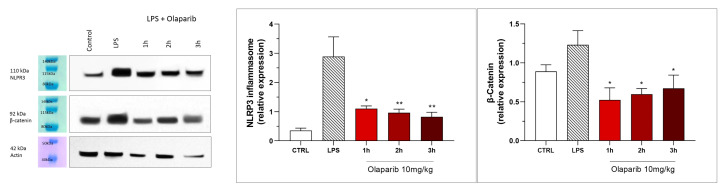
Effect of olaparib post-treatment on NLPR3 inflammasome activation and β-catenin expression in the lung tissue in a murine model of ALI. C57Bl/6 mice received either PBS (30 µL) or LPS (50 µg in 30 µL PBS) by intratracheal instillation in the absence or presence of 10 mg/kg olaparib, delayed to 1, 2, or 3 h post-LPS. After 24 h, lungs were collected, and NLPR3 inflammasome activation and β-catenin expression were measured by Western blotting. * *p* < 0.05 and ** *p* < 0.01 indicate inhibitory effects of olaparib in LPS-treated animals. Data are shown as mean ± SEM, n = 8–12/group.

**Figure 24 cells-11-03789-f024:**
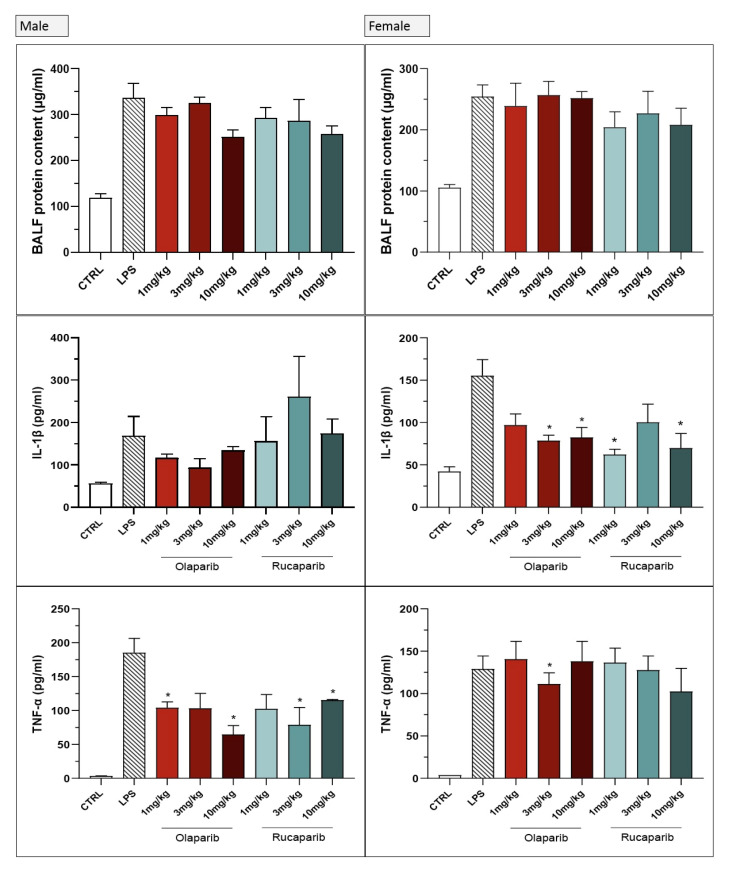
Lack of sex difference in the effect of PARP inhibitors in the LPS-induced ALI model in mice in the pre-treatment paradigm. C57Bl/6 mice received either PBS (30 µL) or LPS (50 µg in 30 µL PBS) by intratracheal instillation in the absence or presence of various doses (1, 3, or 10 mg/kg) of olaparib or rucaparib pre-treatment. After 24 h, bronchoalveolar lavage fluid was collected, and protein content, TNFα, and IL-1β levels were measured. * *p* < 0.05 indicates inhibitory effects of the PARP inhibitors tested in LPS-treated animals. Data are shown as mean ± SEM, n = 4–6/group.

**Figure 25 cells-11-03789-f025:**
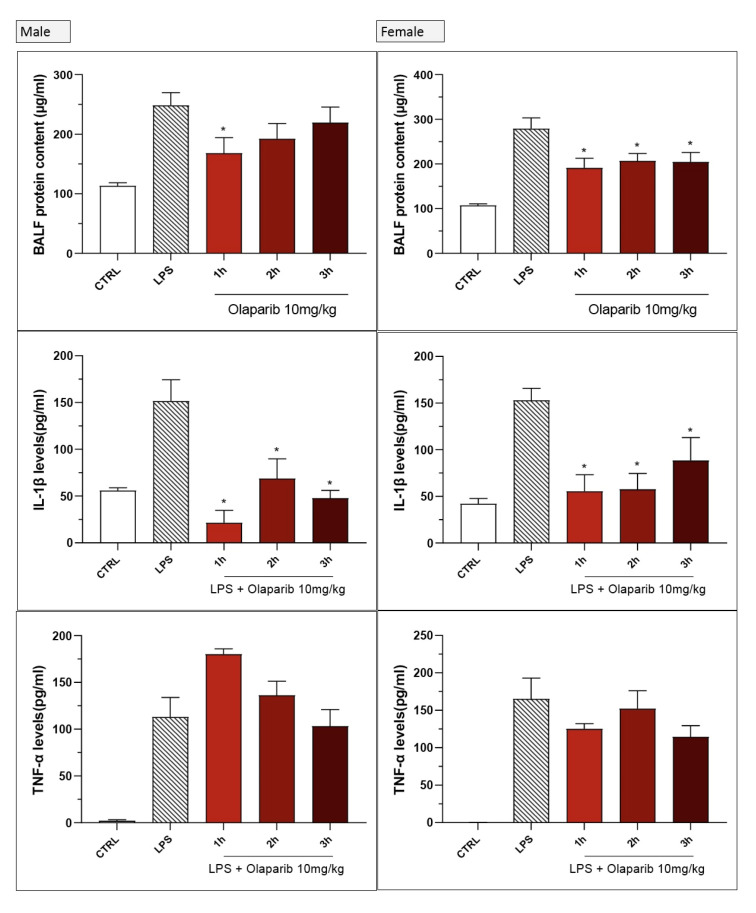
Lack of sex difference in the effect of olaparib in the LPS-induced ALI model in mice in the post-treatment paradigm. C57Bl/6 mice received either PBS (30 µL) or LPS (50 µg in 30 µL PBS) by intratracheal instillation in the absence or presence of 10 mg/kg of olaparib, applied 1, 2, or 3 h after LPS challenge. After 24 h, bronchoalveolar lavage fluid was collected, and protein content, TNFα, and IL-1β levels were measured. * *p* < 0.05 indicates inhibitory effects of olaparib in LPS-treated animals. Data are shown as mean ± SEM, n = 4–6/group.

## Data Availability

The data presented in this study are available on request from the corresponding authors.

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
