# Peer review of "Efficacy of Clinically Used PARP Inhibitors in a Murine Model of Acute Lung Injury"

_cells, 2022, doi:10.3390/cells11233789_

Round 1
Reviewer 1 Report
This study showed the potential for repurposing PARP inhibitors as therapeutics for ALI. However, there may be major corrections and supplements necessary for the manuscript. Detailed comments and suggestion are listed below.
1. Of the 5 inhibitors tested, olaparib showed the best anti-inflammatory effect on ALI. So, what makes olaparib different among the five inhibitors, and what are the mechanisms underlying these differences?
2. In this study, quantitative analysis of inflammatory cytokines and quantitative values of inflammatory cells were shown in tissue analysis to confirm the effect of PARP inhibitors before and after LPS administration. Here, it was confirmed that neutrophils were significantly affected, but mononuclear cells were not uniformly affected by the effects of LPS and PARP inhibitors. What are the reasons for this?
3. In this paper, the effect of olaparib on sex differences was tested. Is there a reason to confirm the effect of gender difference by analyzing only BALF, IL-1β, and TNF-α?
Minor
1. Figure 2,3 => data size too small
2. Figure 4 . The name of the cell line is too small. Also, A and B are not separated.
3. The thickness of the actin band used in Western blot data seems to be inconsistent
4. Line 229 : ')' is written incorrectly
Author Response
Response to Referee 1
- Of the 5 inhibitors tested, olaparib showed the best anti-inflammatory effect on ALI. So, what makes olaparib different among the five inhibitors, and what are the mechanisms underlying these differences? –
Response: Thank you for these comments. We do not necessarily mean to imply that olaparib is the only inhibitor that exerts cytoprotective effects or anti-inflammatory effects. In fact, in the in vitro studies, the effect of all 5 PARP inhibitors tested was comparable against H2O2 induced damage. Part of the reason olaparib was selected for more extensive (in vivo) work is that this PARP inhibitor has the longest clinical history as it was first approved; in addition, this compound, in various preclinical models (e.g. sepsis, burn, various models of neuroinjury and neuroinflammation) has already presented protective effects. Thus, we wanted to test its effect in the current model of ALI, in order to further accumulate data supporting its potential repurposing potential. In the revised manuscript we have made the above points more clear.
- In this study, quantitative analysis of inflammatory cytokines and quantitative values of inflammatory cells were shown in tissue analysis to confirm the effect of PARP inhibitors before and after LPS administration. Here, it was confirmed that neutrophils were significantly affected, but mononuclear cells were not uniformly affected by the effects of LPS and PARP inhibitors. What are the reasons for this?
Response: We must admit that we do not know the exact reason, and this remains to be further studied. It is possible that in the current experimental model PARP plays a more significant role in the regulation of those factors (e.g. soluble mediators and/or cell surface adhesion molecules) that are relevant for neutrophil trafficking, as opposed to those factors that regulate mononuclear cell trafficking. One difference may be important: mononuclear cells have PARP1, while neutrophils lack this enzyme. Thus, the function and trafficking of neutrophils may be mainly influenced through the effects of the PARP inhibitors on other cell types, while the function and trafficking of mononuclear cells may be affected by PARP through a combination of effects on these cells (as well as on their environment). Differential effects of olaparib on neutrophils vs. mononuclear cells has also recently been noted in a model of colitis: in this model neutrophilia was not affected by the PARP inhibitor, while the changes in lymphocyte and monocyte numbers were partially normalized by the PARP inhibitor]. In another study investigating various white blood cell types from myelodysplastic patients, olaparib more significantly affected cells of the myeloid than of the lymphoid lineage. Thus, clearly, olaparib exerts effects on various immune cells in a cell-type- and context-dependent manner. The underlying mechanisms remain to be further investigated. We have included additional discussion in the revision to emphasize these points.
- In this paper, the effect of olaparib on sex differences was tested. Is there a reason to confirm the effect of gender difference by analyzing only BALF, IL-1β, and TNF-α?
Response: We have selected these as these are considered some of the most important ones. Since we have >20 figures, and since none of the other mediators are affected by sex, we did not think we should burden the paper with additional figures. We have made a note that none of the other mediators are different in males vs females.
Minor
- Figure 2,3 => data size too small
Response: In the revision, we have separated these figures to improve visibility.
- Figure 4 . The name of the cell line is too small. Also, A and B are not separated.
Response: We have modified this figure as requested.
- The thickness of the actin band used in Western blot data seems to be inconsistent
Response: We agree with this point on the bottom of the 3 Westerns shown in Fig 9 (now in the revised version Fig 15), but please note that each lane represents an individual animal. Although we have measured protein content prior to loading, the nature of the in vivo experiment (where plasma extravasation occurs, multiple events are happening as part of the disease process that change the levels of various proteins, even potentially constitutive proteins), this type of variability can sometimes happen. Importantly, in these blots, phospho-Act is normalized to total Act (and not actin) and therefore we believe that the data derived from these analyses are correctly interpreted.
- Line 229 : ')' is written incorrectly
Response: We apologize, this was corrected.
In summary, we thank for the comments and suggestions; we believe we have responded to every point raised to satisfy the comments and suggestions. We have a substantial amount of data and information in the manuscript; we hope that you will agree with our revisions. Please also note that the other 2 referees were also generally happy with the material and we have also responded and made all the necessary edits and corrections for those comments
Reviewer 2 Report
please see attached document

Author Response
Response to Referee 2
General comment 1: It needs to be stated that authors have previously published a comprehensive review on the topic of PARP inhibitors in ALI (Szabo, Martins, Liaudet, Am J Respir Cell Mol Biol 63, 571-590), where they nicely summarize previous studies on the same topic. The new manuscript cites this review, but fails to discuss their data in the light of results from several previous studies on PARP inhibitors in LPS-induced ALI. In these older studies probably other drugs of PARP inhibitors might have been used, but comparing the data at least in the discussion is necessary. Otherwise, the novelty of this new study is questionable.
Response: Thank you for the comment. We have written the above review to summarize the field so that we can avoid extensive discussion of the topic in original articles such as the current one. We are quoting in the current manuscript all of the most relevant prior work. Because the focus of the paper is repurposing, in the discussion we are mainly focusing on the effects of clinically used PARP inhibitors. With respect to such agents, the available information in ALI (or, in broader sense, various forms of lung inflammation/injury) is rather limited. In the revised Discussion section of the revised manuscript we are making a more extensive discussion focusing on these studies and comparing them to the current one.
General comment 2: A central issue of this new study is the “safety” i.e. effect on DNA damage. As a method to address this, authors use a TUNEL assay. TUNEL is neither sensitive nor very selective. Meanwhile there are several methods available, that are more suitable to quantify DNA damage. Authors mention this in the discussion as a limitation. For ethical reasons I do not suggest to perform more animal experiments, but cell culture experiments can be adapted using both, H2O2 or LPS, and DNA damage should be addressed in such experiments with a more selective and accurate method.
Response: Thank you for your suggestion. We are afraid this is the only DNA damage assay that we have available in the laboratory currently. We are mentioning this as a limitation, and we are also making a more extensive mention of the sensitivity issue. The limitations part of the manuscript has been expanded including this point.
Introduction: More background information should be given, why PARP inhibitors are promising drugs in the context of ALI and how far has research come in this field? (see your own review article).
Response: Data are steadily accumulating with clinically used PARP inhibitors – mainly olaparib – in various preclinical studies of inflammation and organ injury. These studies should support the concept of repurposing. We have expanded on this issue in the revised manuscript’s discussion. Our group is also currently starting a clinical trial in sepsis with olaparib and we are investigating PARP activation in circulating PBMCs from ARDS patients. But the existence of these studies is not yet public and we cannot include them in this paper.
Material and Methods: The contents of paragraphs 2.4 and 2.12 is largely identical and should be combined.
Response: We have made the necessary correction.
Companies (and maybe even catalog numbers) of used antibodies should be included. Otherwise, dilutions do not mean anything.
Response: We have made the necessary correction.
Paragraph 2.5.: Please provide approval number of animal ethical commission.
Response: We have made the necessary correction.
Figure 1: is hardly readable. Please increase font size.
Response: We have made the necessary correction.
For me, it is not absolutely clear, if there are 12 animals in every group (also with every single dose). How many animals were used for the whole study?
Response: Yes, this was a large study. We had 8-12 animals in each group also with every single dose. We have added the total number of animals in the revised manuscript.
Paragraph 2.9: TUNEL assay: Please describe, how quantification was performed.
Response: We have added more detailed info in the revised manuscript.
Paragraph 2.13: Statistical analysis: It is questionable, whether mean +/- SEM should be used rather than mean +/-SD. What means “at least 2 separate experiments”? Are mean values presented in graphs from a single experiment or from all repeated experiments taken together?
Response: The “2 separate experiments” was a carryover from a previous statistical section and has been corrected. The n number means separate cell cultures or individual animals. The mean represents the mean of the group as specified and the SEM represents the corresponding SEM, as it is used in practically all scientific papers. Regarding SE vs SEM, in many papers SE is used, in others SEM; some groups prefer SE, yet others prefer to express the data using median and confidence intervals… our group typically uses mean and SEM and so far none of the journals we have published in in the past (including PNAS, J Exp Med, and many MDPI journals) had any problems with this approach.
Part1 of the study: in vitro experiment: What were the criteria to choose the U937 and BEAS-2B cell line? Why not also endothelial cells? Primary cells would have been generally preferable, but certainly are not easy to obtain. How were concentrations of H2O2 chosen? Exposure of cell cultures to H2O2 are hardly ever done using full media containing 10% FCS, but serum-free media or very low serum concentrations (0.5%). Serum is a scavenger and has unpredictable effects on the final H2O2 concentration. Also, high concentrations of FCS give a high background in the LDH assay, which makes an accurate quantification of LDH release difficult. The MTT test is rather a read-out for general metabolic activity (due to oxidoreductase enzymes) than only mitochondrial activity.
Response: The Beas2B cells are human non-transformed lung endothelial cells; we have made this clear in the manuscript. We do not have access or protocols for primary cells in our laboratory, sorry. But we have used these two cell types in many prior studies. We are aware of the fact that serum changes the sensitivity of the system to the oxidant. But we have not seen that it makes the experiments less predictable. We think that this is a matter of practice and protocols used in various laboratories. The concentration of the H2O2 was selected based on previous concentration-response studies which we now mention, but did not show because we already have a large number of figures in this manuscript. The LDH assay in our hands works well in these conditions. We have used this type of MTT/LDH approach (in the presence of serum) in several prior studies (e.g. PLoS One. 2015 Jul 28;10(7):e0134227; Br J Pharmacol. 2018 Jan;175(2):284-300; Br J Pharmacol. 2018 Jan;175(2):246-261; Biomolecules. 2020 Mar 13;10(3):447; Biochem Pharmacol. 2020 Dec;182:114267; Shock. 2020 May;53(5):653-665) and previous referees had no major criticism regarding this. We have received 10 days to do a revision of the manuscript, and in this time period we cannot repeat all the experiments without serum. But we do not think that the conclusions would be different if we did them in serum-free medium; only the concentrations of the oxidant would be lower. We have changed the description of the MTT assay as you recommend throughout the manuscript and refer to “MTT-converting activity” rather than “mitochondrial activity”.
Results: Figure 2 is not readable at all. Even when magnified, it is blurry due to low resolution. As far as I can see, the labeling is also not conclusive. Headings should not only include the type of cell line, but also which test (MTT/LDH) and what time point (2h/24h). All other details cannot be assessed, as they are not legible. The same applies to Figure 3.
Response: We have broken out these figures into several ones and now they are more readable.
Figure 4: DNA damage must be assessed with an additional, more selective and accurate method. If TUNEL results remain in the manuscript, microscopy images staining should be shown. How was DNA damage quantified after TUNEL? Control values in 600 uM and 30 uM H2O2 panel varies. It seems, that there is no difference in % damage between low and high H2O2 concentration, which is surprising.
Response: This was done using an ELISA based method, not microscopy. We have added further details to the methods. But because the method is ultimately based on absorbance values that we obtain from a plate reader, we cannot add images. The concentration of the oxidant was different for the two cell types, because the sensitivity of the two cell types was so different, as this was already shown in prior figures (and this is now also better explained in the Methods).
Figure 6: Stars of significance levels are missing. See also sentence in lines 255-257. Is this correct?
Response: The significances are only showing the effect of the PARP inhibitors in the presence of LPS, vs. LPS. (It is obvious that the LPS increases these values over the control and this is not necessary to show). In some parts of the figure the PARP inhibitor+LPS vs LPS alone is not significantly different from each other and it is therefore not shown. We have also checked the lines specified above and edited as necessary.
Figure 7: TUNEL staining can be kept for animal studies. Please provide detailed information on the procedure. Please show microscopy image of TUNEL stain.
Response: We have modified the figure (now it is Fig 13/14 in the revised manuscript and added representative microscopy images to include a new figure, Fig 13)
Figure 8: It seems, that indicators of levels of significance are missing in the bar charts.
Response: The significances are only showing the effect of the PARP inhibitors in the presence of LPS, vs. LPS.
Figure 9: All labeling needs to be larger, as it is currently hardly legible. Here expression levels of p65 and caspase 3 are quantified by Westernblot. More conclusive is quantifying activation of NF-KB (activated/phosphorylated intermediates of the pathway: see Meier-Soelch et al. Cancers 2021, 13,5354) and caspase-3 (=cleaved caspase 3). See also line 357: The text says: “The inhibition of NF-KB activation….” : this is incorrect, as it has not been analyzed. The Westernblot image of Akt is weird: Some bands of Akt-1 staining are missing. If these values are really so low, the ratio of p-Akt/Akt must be extremely high (this is not the case in the bar chart on the right). Which phosphorylated form of Akt was analyzed (T308 or S473?) Same holds true for Figure 10: activation of NLRP3 should be analyzed.
Response: We agree with this point on the bottom of the 3 Westerns shown in Fig 9 (now in the revised version Fig 15), but please note that each lane represents an individual animal. Inherently, animal tissue homogenate-based Westerns tend to have a higher animal-to-animal variability than replicates of cell homogenates, for example – especially when the animals are also subjected to an injurious agent such as LPS. Although we have measured protein content prior to loading, the nature of the in vivo experiment (where plasma extravasation occurs, multiple events are happening as part of the disease process that change the levels of various proteins, even potentially constitutive proteins), this type of variability can sometimes happen. Importantly, in these blots, phospho-Act is normalized to total Act (and not actin) and therefore we believe that the data derived from these analyses are correctly interpreted. We have corrected the part about NfKB ‘activation’ (we only measure a key components of the complex and not actual activation, but at the same time the 65-kDa subunit of NF-kappa B is well established to function as a potent transcriptional activator and a target for v-Rel-mediated repression and if this subunit is lower, it is likely that the NFkB system’s activity is suppressed). Regarding NLRP3, we are measuring the expression of the protein but it is well established in the literature that its expression level correlates well with the activity of this pathway. Regarding conducting additional analysis: we are afraid that we do not have any samples left to do more work. In the time given for the revision we cannot go back and repeat the entire experiment, we apologize for that.
Figure 13: Please show microscopy image of TUNEL.
Response: We have added representative microscopy images for the earlier figure; this study is only the post-treatment part and we have so many figures and subfigures that we don’t think adding it here again would add a lot to the paper.
Figure 14: Indicators of levels of significance seem to be missing in some cases
Response: The significances are only showing the effect of the PARP inhibitors in the presence of LPS, vs. LPS.
Figure 15: Same applies to NF-KB and caspase as above. AKT-Blot is weird: There are more actin bands in the loading control lane compared to the pAkt and Akt lanes. Why was the ratio of ERK1/ERK2 calculated, and not p-ERK/ERK? Are there indicators of significance missing in the p-Akt/Akt bar chart?
Response: Regarding the Akt blots, please see our previous response. Regarding the ERK1/ERK2 ratio, we are aware of the fact that most papers study the phosphorylated forms. However, a direct relationship between the ratio of phosphorylated ERKs and the quantitative expression ratio of ERKs has been previously shown (e.g. Lefloch et al., 2008), therefore, to evaluate the ratio between ERK1 and ERK2 in a biological sample, one can either measure the ERK1/2 ratio or the ratio between the dually-phosphorylated ERK isoforms. We have added this as a limitation of the study to the discussion.
Figure 16: The importance of ß-catenin expression is not explained.
Response: We have expanded the discussion on this.
Figure 17: Male/female groups: Why is there n=4-6/group? Originally, there were 12 animals and 50% of both gender? (same in Figure 18)
Response: We have used mixed groups of animals (n=8-12 per treatment group) but in some cases, the ratio was not exactly 50%. The legend has been corrected to state 4-8 animals/groups: in rare cases we had 4 males and 8 females; in most cases we had 6 and 6 or 5 and 7 in the groups that had 12 animals. In the groups that had 8 animals in total, we had 4 males and 4 females.
In summary, we thank for the comments and suggestions; we have done the best we could given our available methodologies and resources to satisfy the comments and suggestions. We have a substantial amount of data and information in the manuscript; we hope that you will agree with our revisions. Please also note that the other 2 referees were also generally happy with the material and we have also responded and made all the necessary edits and corrections for those comments.
Reviewer 3 Report
1) All western blot images are extensively cropped and need to be more generously cropped before submission to include the molecular weight markers above and below the protein band of interest.
2) All the graph axes labels are too small in figures to be legible (Figures 2 and 3). The graphs included in the figures should be sized appropriately to be readable.
3) Histological images are missing the scale bars.
4) Provide a comprehensive model for the hypothesis and the results presented in the manuscript. This could explain the mechanism of action.
5) Please discuss the caveats associated with the study in greater detail.
Author Response
Response to Referee 3
1) All western blot images are extensively cropped and need to be more generously cropped before submission to include the molecular weight markers above and below the protein band of interest We have redone these figures as suggested.
2) All the graph axes labels are too small in figures to be legible (Figures 2 and 3). The graphs included in the figures should be sized appropriately to be readable. In the revision, we have separated these figures to improve visibility.
3) Histological images are missing the scale bars. In the revision, we have included a scale bar.
4) Provide a comprehensive model for the hypothesis and the results presented in the manuscript. This could explain the mechanism of action. – In the revised manuscript we have re-emphasized the likely mechanisms in the final section of the Discussion (Conclusion section).
5) Please discuss the caveats associated with the study in greater detail. – In the revised manuscript we have expanded on the limitations/caveats section in the Discussion.
In summary, we thank for the comments and suggestions; we believe we have responded to every point raised to satisfy the comments and suggestions. We have a substantial amount of data and information in the manuscript; we hope that you will agree with our revisions. Please also note that the other 2 referees were also generally happy with the material and we have also responded and made all the necessary edits and corrections for those comments.
Round 2
Reviewer 2 Report
The manuscript is improved.
Minor: spelling error in line 198: PVDF instead of PDVF
Reviewer 3 Report
The authors have addressed the concerns raised in the first round of reviews. The manuscript is significantly improved and reads much better. Overall, the manuscript presents an interesting question important to the field in general and the presented data is logically presented to validate the hypothesis. I recommend the manuscript for publication.